**Data Availability Statement:** All relevant data are within the paper.

**Funding:** This article´s publication charges are supported by a grant from the Suzanne and Hans

# Attitudes of medical students towards interprofessional education: A mixed-methods study

**Joana Berger-Estilita**[1]*, **Hsin Chiang**[1], **Daniel Stricker**[2], **Alexander Fuchs**[1], **Robert Greif**[1,3], **Sean McAleer**[4]

**1** Department of Anaesthesiology and Pain Medicine, Inselspital, Bern University Hospital, University of Bern, Bern, Switzerland, **2** Institute for Medical Education, University of Bern, Bern, Switzerland, **3** School of Medicine, Sigmund Freud University Vienna, Vienna, Austria, **4** Centre for Medical Education, University of Dundee, Dundee, United Kingdom

* joana.berger-estilita@insel.ch

## Abstract

### Background

Interprofessional Education (IPE) aims to improve students' attitudes towards collaboration, teamwork, and leads to improved patient care upon graduation. However, the best time to introduce IPE into the undergraduate curriculum is still under debate.

### Methods

We used a mixed-methods design based on a sequential explanatory model. Medical students from all six years at the University of Bern, Switzerland (n = 683) completed an online survey about attitudes towards interprofessional learning using a scale validated for German speakers (G-IPAS). Thirty-one medical students participated in nine semi-structured interviews focusing on their experience in interprofessional learning and on the possible impact it might have on their professional development.

### Results

Women showed better attitudes in the G-IPAS across all years ($p$ = 0,007). Pre-clinical students showed more positive attitudes towards IPE [Year 1 to Year 3 ($p$ = 0.011)]. Students correctly defined IPE and its core dimensions. They appealed for more organized IPE interventions throughout the curriculum. Students also acknowledged the relevance of IPE for their future professional performance.

### Conclusions

These findings support an early introduction of IPE into the medical curriculum. Although students realise that interprofessional learning is fundamental to high-quality patient care, there are still obstacles and stereotypes to overcome.

Biäsch Foundation for Applied Psychology (Nr. 2020-23). The funders had no role in study design, data collection and analysis, decision to publish, or preparation of the manuscript.

**Competing interests:** RG is the Board Director of Training and Education for the European Resuscitation Council, the Task Force Chair Education, Implementation, and Team of ILCOR, and member of the direction of the MME Programme of the University of Bern. SM is the Programme Director and Senior Lecturer at the Centre for Medical Education, University of Dundee. This does not alter our adherence to PLOS ONE policies on sharing data and materials. The remaining authors report no competing interests. We confirm that this manuscript is not under consideration by another journal. It is our own work and was not sponsored by the industry.

## Trial registration

ISRCTN 41715934.

## Introduction

The World Health Organization (WHO) defines Interprofessional Education (IPE) as, when "students from two or more professions learn about, from, and with each other to enable effective collaboration and improve the quality of care" [1]. Evidence shows that interprofessional (IP) healthcare interventions improve patient outcomes, such as higher medication safety or reduced length of hospital stay [2] by enhancing the communication and interpersonal skills of healthcare professionals, as well as their collaboration and teamwork skills [3]. The Interprofessional Collaborative Practice (IPEC) outlines IPE's core competencies which concentrate on four main domains: Ethics & Values, Roles & Responsibilities, IP Communication and Teamwork [4].

Nevertheless, the complexity of teaching for different healthcare disciplines, logistical problems and busy timetables raise issues concerning the introduction of IPE interventions. Current undergraduate literature shows a trend for earlier IPE introduction [5, 6], but the optimal timing for the IPE intervention is unclear [7].

IPE interventions can be measured by using validated attitudes scales based on IPE domains. Until recently, only a few conceptual tools for assessing attitudes towards IPE existed [8]. The Readiness for Interprofessional Learning Scale (RIPLS) [9] and the extended RIPLS [10] are common examples. Unfortunately, many scales were developed before the IPEC report, and do not integrate all four recommended core competencies [11]. The Interprofessional Attitudes Scale (IPAS) [12]–developed and validated in 2015—uses items from the extended RIPLS and new items to embody all four IPEC domains. This scale has been validated for German speakers [13].

The Medical Faculty of the University of Bern (UniBe) is one of the largest in Switzerland with about 1500 students. The study of Medicine starts with a 3-year bachelors programme focusing on basic science (e.g. physics, chemistry, biology, physiology, biochemistry and anatomy) followed by a 3-year masters programme with a strong practical focus, composed mostly of small group interactions (problem-based learning) and clinical clerkships [14]. Since 2010 the medical faculty and nursing schools have been offering optional two half-day interprofessional internships for their students in the first and third semesters. Further interprofessional activities include a compulsory seminar on confidentiality in cooperation with the Bern University of Applied Sciences and the Institute for Medical Education of the University of Bern (UniBe) as well as the compulsory Intravenous Cannulation course, both taught in the first academic year, during which the learning groups and the team of peer tutors are interprofessionally allocated.

The aims of this study are: (1) to determine whether there are changes in attitudes towards interprofessionality between the bachelors (pre-clinical) and masters (clinical) programme of the curriculum by using a validated attitudes scale, and (2) to ascertain the ideal time in the medical curriculum to introduce IPE interventions.

## Materials and methods

We used a sequential qualitative-quantitative mixed methods design [15]. The quantitative cross-sectional survey collected students' demographic data and included all 24 items of the

German Interprofessional Attitudes Scale G-IPAS [13] using an online platform (SurveyMonkey Inc, San Mateo, California, USA). Semi-structured interviews explored individual students' experiences with IPE interventions, and the impact they had on their professional development. All medical students actively enrolled in the Faculty of Medicine of the University of Bern, Switzerland, during the academic year 2019/2020 were eligible for inclusion in the study. The study was conducted in German.

## Ethical considerations

The participants gave written informed consent and the Bern Cantonal Ethics Committee (Req-2019-00743, 23.08.2019) waived the need for ethics approval. The survey link included a covering letter reiterating the goals of the study and "consent by participation" was obtained [16]. We used ID numbers to code students and requested no identifying data. Data was stored in a secure repository accessible to the investigators only. All procedures from this investigation followed the Helsinki Declaration [17]. All researchers complied with the Data Protection Act [18] and the Swiss Law for Human Research [19]. This study was registered with the number ISRCTN41715934.

## Procedure

Students received an e-mail from the Medical Faculty deanery in October 2019 with the link to the online G-IPAS survey via the online platform. The survey was open from 7th October to 15th December 2019, and two reminders were sent.

The German Interprofessional Attitudes Scale is a 24-item questionnaire with 3 subscales ("*Teamwork, Roles and Responsibilities*", "*Patient-centeredness*" and "*Healthcare Provision*"). Participants had to answer the questions using a Likert scale with 1 representing "Strongly Disagree", 2 "Disagree", 3 "Neutral", 4 "Agree" and 5 "Strongly Agree". The G-IPAS has been shown to be a reliable instrument, representative of the original American IPAS dimensions [12] and it has been translated, culturally adapted and validated in German-speaking countries for the assessment of interprofessional attitudes [13].

After completion of the online G-IPAS questionnaire, students were invited to participate in nine semi-structured interviews, which took place at the Department of Anaesthesiology and Pain Therapy, Inselspital, Bern, Switzerland in November 2019. An interview guide was used to conduct the one-hour session. Students provided demographic data (e.g. age, year of studies) and were asked about their understanding of IPE and the (dis)advantages of this type of teaching strategy. We discussed the survey results and asked their opinion on optimal IPE interventions (duration, format and content). Data was audio- and video recorded.

## Sampling

For the quantitative phase, we used a non-probability convenience sample and included all medical students from the Bern Faculty of Medicine enrolled in the academic year 2019/2020 (n = 1550). We aimed to include 100 students for each year, and at least 600 students overall, following recommendations for sample size survey research [20]. As the study was sequential in nature, it was impossible to pre-emptively select participants for the qualitative phase. We used purposive sampling for the nine semi-structured interview groups.

## Data analysis

We performed a descriptive analysis of the survey data with sub-group analysis per year of studies. Global scale, dimensions, and individual items were assessed for normal distribution

with the Shapiro-Wilks test and visual assessment of residuals and Q-Q Plots. Two-way analysis of variance (ANOVA) with gender and the stratified study years (year 1 to 6) as between subjects' factors were conducted separately for the means of all subscores as well as the mean overall G-IPAS score as dependent variables. Separate independent samples t-tests were conducted for the between subjects' factor previous experience in healthcare and having parents working in the healthcare system for the overall G-IPAS score, with correction for multiple testing. Additionally, an independent samples t-test was conducted to compare the overall G-IPAS score in pre-clinical (years 1–3) and clinical years (years 4–6). Quantitative data was analysed with SPSS v26 (IBM, New York, USA).

Because the G-IPAS has only recently been introduced, we decided to perform an additional confirmatory analysis of its validity and reliability. For survey validity, we used a factor analysis using the Scree test for factor extraction and Varimax rotation with Kaiser-normalization. Data was assessed for factorability with Bartlett´s test of sphericity, and the Kaiser-Meyer-Olkin (KMO) measure of sampling adequacy. For reliability, Cronbach's alpha was determined. Cronbach's alpha should be at least of 0.7 for the instrument to be considered reliable [21].

Data from the semi-structured interviews was processed according to the Miles and Huberman [22] framework for data analysis: data segmenting, editing and summarizing, followed by data display, and finally conclusion verification. HC transcribed all interviews. JBE and HC corrected and verified transcriptions of the interviews and we sent summaries of the interview to each participant as a form of respondent validation [23]. JBE and HC both coded the first group interview independently using the software MaxQDA2020® (Verbi, Berlin, Germany) and agreed on the coding scheme for the remaining interviews. Memoing was performed parallel to coding. All interviews were coded in a phased fashion, with interim analysis, to check for saturation.

Direct quotations from the interviews were translated into English using a functionalist approach of *creation of equivalent translation structures* as described by Enzenhofer and Resch [24]. One author (HC, German-speaking) translated the citations from German to English *ipsis verbis* with the aid of an online tool (Google Translate®). The second author (SM, English-speaking), performed changes to ensure that the target text could be understood by the reader.

## Results

### Quantitative analysis

Six-hundred and seventy-seven students replied to the online survey (response rate: 43,7%). Incomplete questionnaires (n = 111) were excluded and 4 students did not report year of studies. We included 562 completed questionnaires in the final analysis.

### Confirmatory analysis of the instrument's validity and reliability

The initial three-factor model (Teamwork, Roles & Responsibilities, Patient-centeredness and Healthcare Provision) explained 48% of the total variance. After rotation, a simple structure with loadings on to the three components emerged. This is consistent with previous research [13]. The calculated Cronbach's alpha for G-IPAS was 0.855.

### Demographic characteristics

Participants' demographics are shown in Table 1. 54% of the students reported previous experience as healthcare providers and over 80% of participants were Swiss German. Most frequent IPEs mentioned were the Intravenous Cannulation course (n = 125), the Confidentiality seminar (n = 98), and the optional interprofessional rotation (n = 43).

**Table 1. Participant´s demographics for the quantitative data.**

| Year of studies | Year 1 (n = 74) | Year 2 (n = 84) | Year 3 (n = 108) | Year 4 (n = 93) | Year 5 (n = 103) | Year 6 (n = 100) | Total (n = 562) |
|---|---|---|---|---|---|---|---|
| **Women [n(%)]** | 50 (68) | 56 (67) | 71 (66) | 68 (66) | 71 (69) | 63 (63) | 379 (67) |
| **Age (mean ± SD)** | 20.5±2.4 | 21.1±2.0 | 22.6±3.4 | 23.4±2.6 | 24.1±2.0 | 25.6±2.0 | 23.1±3.0 |
| **Previous IPE interventions [n (%)]** | | | | | | | |
| None | 69 (95) | 38 (45) | 38 (35) | 25 (27) | 57 (55) | 60 (60) | 287 (51) |
| $\leq$ 2 courses | 2 (3) | 44 (52) | 64 (59) | 55 (60) | 40 (39) | 31(31) | 236 (42) |
| > 2 courses | 2 (3) | 2 (2) | 6 (6) | 12 (13) | 6 (6) | 8 (8) | 36 (6) |
| **Previous experience in healthcare [n (%)]** | | | | | | | |
| yes | 31(42) | 51 (61) | 62 (56) | 37 (40) | 60 (58) | 60 (60) | 301 (54) |
| **Parents working in the healthcare system [n(%)]** | | | | | | | |
| yes | 25 (34) | 26 (31) | 32 (30) | 41 (44) | 44 (43) | 34 (34) | 202 (36) |

## German interprofessional attitudes scale questionnaire

Table 2 shows the mean scores of each G-IPAS item. Five of the nine items in the subscale "Teamwork, Roles and Responsibilities", six of the eight in "Patient-Centeredness" and one in "Health Provision" were significantly higher in females. In the subscale analysis, only "Teamwork, Roles and Responsibilities" decreased significantly with an increase in study years (p<0.001). Males showed lower mean scores in the subscale "Teamwork, Roles and Responsibilities" (p = 0.002) and "Patient-centeredness" (p<0.001) but not in the subscale "Health Provision" (Table 3).

The two-way ANOVA of the G-IPAS mean score showed a statistically significant main effect for gender (F(1, 550) = 7.129, p = 0.008, $\eta^2_p$ = 0.013), with women achieving overall higher mean GIPAS scores. The main effect of study year (F(5, 550) = 2.109, p = 0.063, $\eta^2_p$ = 0.019) and the interaction effect between gender and study year (F(5, 550) = 1.927, p = 0.088, $\eta^2_p$ = 0.017) was not statistically significant. The independent samples t-tests showed no statistically significant differences for previous experience in healthcare and having parents working in the healthcare system.

An independent samples t-test revealed a significant difference in the means of the overall G-IPAS score between pre-clinical (M = 4.22, SD = 0.40) and clinical years (M = 4.13, SD = 0.40) (p = 0.007).

## Qualitative analysis

We performed nine group interviews (maximum of 4 students each), 31 participants in total. All study years were represented [Year 1: n = 5 (16%), Year 2: n = 8 (26%), Year 3: n = 2 (7%), Year 4: n = 8 (26%), Year 5: n = 7 (23%), Year 6: n = 1 (3%)]. There were 20 female students (64,5%), 16 (51.6%) students had previous experience in healthcare work, 24 students (77.4%) had at least one parent working in healthcare, and 19 students (61.3%) had healthcare professionals as close friends. Table 4 depicts the overall frequency of the quotes in each of the interviews. There were similar distributions of codes across genders and years of study.

Three main categories emerged from the focus groups: a) *awareness of IPE*, b) *barriers to IPE*, and c) *expectations of IPE*.

### a) Awareness of IPE

**Definition of interprofessional education.** The interviews demonstrated that students could correctly define IPE, as per the WHO definition [1] (Table 5, Quote 1). Learning opportunities appeared when topics overlap and are relevant for the healthcare groups involved.

**Table 2. Mean values for G-IPAS individual components.**

| Item[a] | German Interprofessional Attitudes Scale (G-IPAS) (n = 562) | Women | Men | Total | *p* value |
|---|---|---|---|---|---|
| | **Teamwork, roles and responsibilities [Mean(SD)]** | | | | |
| TFV1 | *Shared learning before graduation will help me become a better team worker* | 3.79 (1.01) | 3.55 (1.13) | 3.71 (1.05) | **0.015** |
| TFV2 | *Shared learning will help me think positively about other professionals* | 3.33 (1.09) | 3.14 (1.18) | 3.27 (1.12) | 0.059 |
| TFV3 | *Learning with other students will help me become a more effective member of a health care team.* | 3.91 (1.01) | 3.58 (1.20) | 3.81 (1.08) | **0.001** |
| TFV4 | *Shared learning with other health sciences students will increase my ability to understand clinical problems.* | 3.30 (1.05) | 3.12 (1.09) | 3.24 (1.06) | 0.061 |
| TFV5 | *Patients would ultimately benefit if health sciences students worked together to solve patient problems.* | 4.20 (0.93) | 3.96 (0.92) | 4.12 (0.93) | **0.004** |
| TFV6 | *Shared learning with other health sciences students will help me communicate better with patients and other professionals.* | 4.03 (0.10) | 3.69 (1.11) | 3.92 (1.05) | **0.000** |
| TFV7 | *I would welcome the opportunity to work on small group projects with other health sciences students.* | 3.48 (1.18) | 3.43 (1.23) | 3.47 (1,19) | 0.644 |
| TFV8[c] | *It is not necessary for health sciences students to learn together* | 3.72 (1.07) | 3.34 (1.28) | 2.4 (1.15) | **0.001** |
| TFV9 | *Shared learning will help me understand my own limitations* | 3.23 (1.11) | 3.29 (1.14) | 3.25 (1.12) | 0.550 |
| | **Patient-centeredness [Mean(SD)]** | | | | |
| PZ1 | *Establishing trust with my patients is important to me* | 4.90 (0.31) | 4.81 (0.40) | 4.88 (0.34) | **0.008** |
| PZ2 | *It is important for me to communicate compassion to my patients* | 4.87 (0.39) | 4.71 (0.50) | 4.81 (0.43) | **0.000** |
| PZ3 | *Thinking about the patient as a person is important in getting treatment right* | 4.75 (0.50) | 4.59 (0.59) | 4.70 (0.53) | **0.002** |
| PZ4 | *In my profession, one needs skills in interacting and cooperating with patients* | 4.88 (0.39) | 4.83 (0.45) | 4.86 (0.41) | 0.166 |
| PZ5 | *It is important for me to understand the patient's side of the problem* | 4.80 (0.46) | 4.68 (0.56) | 4.76 (0.50) | **0.018** |
| PZ6 | *It is important for health professionals to understand what it takes to effectively communicate across cultures* | 4.66 (0.53) | 4.52 (0.68) | 4.62 (0.59) | **0.017** |
| PZ7 | *It is important for health professionals to respect the dignity and privacy of patients while maintaining confidentiality in the delivery of team-based care* | 4.81 (0.42) | 4.75 (0.53) | 4.79 (0.46) | 0.172 |
| PZ8 | *It is important for health professionals to provide excellent treatment to patients regardless of their background (e.g., race, ethnicity, gender, sexual orientation, religion, class, national origin, immigration status, or ability)* | 4.95 (0.22) | 4.89 (0.38) | 4.93 (0.28) | **0.035** |
| | **Healthcare Provision [Mean(SD)]** | | | | |
| GHV1 | *It is important for health professionals to work with public health administrators and policy makers to improve delivery of health care* | 4.07 (0.78) | 4.20 (0.88) | 4.11 (0.82) | 0.069 |
| GHV2 | *It is important for health professionals to work on projects to promote community and public health* | 4.14 (0.80) | 4.17 (0.86) | 4.15 (0.82) | 0.684 |
| GHV3 | *It is important for health professionals to work with the legislators to develop laws, regulations, and policies that improve health care* | 4.07 (0.82) | 4.28 (0.76) | 4.14 (0.80) | **0.002** |
| GHV4 | *It is important for health professionals to work with non-clinicians to deliver more effective health care.* | 4.06 (0.84) | 4.09 (0.97) | 4.07 (0.88) | 0.737 |
| GHV5 | *It is important for health professionals to focus on populations and communities, in addition to individual patients, to deliver effective health care* | 4.02 (0.87) | 4.18 (0.86) | 4.07 (0.87) | 0.052 |
| GHV6 | *It is important for health professionals to be advocates for the health of patients and communities* | 4.16 (0.85) | 4.23 (0.90) | 4.19 (0.87) | 0.343 |
| GHV7 | *It is important for health professionals to respect the unique cultures, values, roles/responsibilities, and expertise of other health professions* | 4.65 (0.56) | 4.56 (0.59) | 4.62 (0.57) | 0.071 |

[a]The items have been translated from the German language. TFV = Teamwork, roles and responsibilities, PZ = Patient-centredness, GHV = Health Provision.

Table 3. Mean scores for the G-IPAS score and subscale scores, stratified by gender and year of studies.

| | Year 1 | | Year 2 | | Year 3 | | Year 4 | | Year 5 | | Year 6 | | Overall average | | p value |
|---|---|---|---|---|---|---|---|---|---|---|---|---|---|---|---|
| **Overall Scores** | Women (n = 50) | Men (n = 24) | Women (n = 56) | Men (n = 28) | Women (n = 71) | Men (n = 37) | Women (n = 68) | Men (n = 25) | Women (n = 71) | Men (n = 32) | Women (n = 63) | Men (n = 37) | Women (n = 379) | Men (n = 183) | |
| **Teamwork, roles and responsibilities [Mean (SD)]** | 3.95 (0.59) | 3.70 (0.63) | 3.63 (0.80) | 3.74 (0.81) | 3.73 (0.52) | 3.57 (0.61) | 3.61 (0.74) | 3.08 (0.90) | 3.60 (0.75) | 3.47 (0.82) | 3.5 (0.83) | 3.20 (1.05) | 3.67 (0.72) | 3.46 (0.84) | **0.002** |
| | **3.87 (0.61)** | | **3.67 (0.81)** | | **3.68 (0.56)** | | **3.47 (0.81)** | | **3.56 (0.77)** | | **3.39 (0.92)** | | **3.60 (0.77)** | | |
| **Patient-centeredness [Mean(SD)]** | 4.85 (0.2) | 4.58 (0.40) | 4.76 (0.31) | 4.78 (0,33) | 4.82 (0.24) | 4.75 (0.34) | 4.83 (0.2) | 4.70 (0.26) | 4.85 (0.22) | 4.76 (0.32) | 4.83 (0.22) | 4.72 (0.35) | 4.83 (0.23) | 4.72 (0.33) | **0.000** |
| | **4.76 (0.31)** | | **4.76 (0.32)** | | **4.80 (0.28)** | | **4.79 (0.23)** | | **4.82 (0.25)** | | **4.79 (0.29)** | | **4.79 (0,28)** | | |
| **Healthcare Provision [Mean(SD)]** | 4.33 (0.54) | 3.99 (0.70) | 4.08 (0.70) | 4.31 (0.56) | 4.20 (0.48) | 4.26 (0.57) | 4.08 (0.53) | 4.34 (0.49) | 4.18 (0.55) | 4.23 (0.61) | 4.14 (0.59) | 4.28 (0.55) | 4.17 (0.57) | 4.24 (0.58) | 0.207 |
| | **4.22 (0.61)** | | **4.16 (0.67)** | | **4.22 (0.52)** | | **4.16 (0.54)** | | **4.02 (0.57)** | | **4.19 (0.57)** | | **4.19 (0.57)** | | |
| **Overall G-IPAS [Mean (SD)]** | 4.36 (0.35) | 4.08 (0.48) | 4.14 (0.48) | 4.25 (0.46) | 4.23 (0.29) | 4.16 (0.39) | 4.16 (0.38) | 3.99 (0.38) | 4.19 (0.36) | 4.13 (0.36) | 4.07 (0.35) | 4.01 (0.38) | 4.11 (0.44) | 4.20 (0.40) | **0.008** |
| | **4.27 (0.41)** | | **4.17 (0.48)** | | **4.12 (0.32)** | | **4.11 (0.38)** | | **4.11 (0.38)** | | **4.05 (0.36)** | | **4.12 (0.36)** | | |

P-values indicate the significance of the main effect gender for the overall average Scores obtained from the separate ANOVAs.

Such interventions allow for exchange of knowledge or skills and sharing of different experiences, which improves understanding and communication between groups, and builds trust. IPE can refer to learning about the roles, responsibilities, competencies and duties of other healthcare professionals (Table 5, Quotes 2 and 3). It was also noted that IPE benefits patient care and helps build a social network of people within the working environment (Table 5, Quote 4).

**Recognition of interprofessional education in the medical curriculum.** The most vividly recalled experience was the intravenous cannulation workshop, currently being taught during the first year of studies. The course was considered interprofessional because it was taught by a registered nurse and held in a small-group workshop, with groups of up to six students (including nurses, midwives and sometimes pre-hospital technicians). All participants mentioned that it was a positive experience and that they profited from the course. Main positive aspects mentioned included: (1) the teaching and then the practice with a skilled nursing student; (2) the relaxed, informal interaction; and (3) the exchange of information and guidance from the nursing students, with tips from daily practice.

*"I could even benefit a lot from the nursing students or the midwives. You really noticed that they already did it on real people when we were still practicing on the models. And they already had routine and could give us good practical advice."* (Interview 2, Student 2)

*"The intravenous cannulation (. . .) was shown by the nursing student and not by the course instructor, who was a medical student in the higher year because he simply said that the nurse could do it better and had more experience. I thought that was extremely good, that he then said that she could do better and should show it."* (Interview 7, Student 2)

*"(. . .) we were deliberately divided into my group, that you were always with someone who was not a medical student, which I found very exciting."* (Interview 5, Student 3)

*"I could benefit [from the intravenous cannulation course] because we had a qualified nurse (. . .), who could actually show me how it worked, better than the instructor. And otherwise, it was a relaxed atmosphere."* (Interview 8, Student 2)

However, most students realised that nursing students already had the given competency and were bored/frustrated during the workshop. Some medical students observed other peers

**Table 4. Coding frequency across all interviews.**

|  | Frequency (n) | Percentage (%) |
|---|---|---|
| **Participant´s age** | 31 | 3.33 |
| **Participant´s year of studies** | 31 | 3.33 |
| **Participant´s previous work experience** | 15 | 1.61 |
| **Participant´s ties with healthcare** | 41 | 4.40 |
| **Comments on filling the GIPAS form** | 25 | 2.68 |
| **Definition of IPE** | 44 | 4.72 |
| **Goals of IPE** | 48 | 5.15 |
| **Advantages of IPE** | 112 | 12.02 |
| **Disadvantages of IPE** | 101 | 10.84 |
| **Examples of IPE during medical course** | 96 | 10.30 |
| **Attitudes towards IPE** | 63 | 6.76 |
| **Attitudes: Absence of IPE** | 34 | 3.65 |
| **Examples of wished for interventions** | 70 | 7.51 |
| **Desired format of the IPE course** | 71 | 7.62 |
| **Desired Year of studies for IPE** | 92 | 9.87 |
| **Desired Frequency of IPE** | 50 | 5.36 |
| **Ideal group size for IPE interventions** | 8 | 0.86 |
| **Total number of coded citations** | 932 | 100 |

having discriminating attitudes towards nursing students. Most were unhappy to be in a workshop where they knew less than their nursing counterparts and could not contribute to any exchange in knowledge.

> "[During the intravenous cannulation course] I heard from many nursing students that they didn't understand that they were doing there. They could already do it and had clinical experience. It was therefore unnecessary for them to take the course and a waste of time" (Interview 5, Student 1)

> "I noticed that a colleague of mine got upset about the teaching at the intravenous cannulation course and mentioned that "she is just a nurse anyway". I then asked him directly, "that means that she can do less?" And he answered "yes" and stood by it. He really meant it, and only because the nurse had other competencies. And he was a first-year student." (Interview 7, Student 1)

**Table 5. Subcategory "definition of IPE" elements and representative cites.**

| Subtheme with explanation | Representative cites (exemplary) from semi-structured interviews |
|---|---|
| ***Definition of IPE***<br>Learning that occurs with 2 or more different health professionals or healthcare students<br>• about each other´s professions<br>• with other professions about a common topic<br>• to enable effective collaboration<br>• to improve patient outcomes | Quote 1, Interview 8, Student 3: "(. . .) at least 1 person from a different professional group is present as a medical student."<br>Quote 2, Interview 1, Student 2: "I can only agree with the keyword "more efficient cooperation". I think it is all about having the knowledge and understanding, what are the tasks, the competencies of another team member and how can you support and benefit from each other."<br>Quote 3, Interview 1, Student 3: "Who does which tasks–it is important that you learn that, so that you focus on the patient."<br>Quote 4, Interview 6, Student 3: (. . .) so that people who work in the health sector optimally form a network with each other and work effectively together." |

*"I don't know what the others should learn from us. We can't do anything! Maybe we know more, but that doesn't interest them that deeply either."* (Interview 2, Student 2)

It was also noted that if groups were not deliberately mixed, students from the same profession tended to group together and quality learning was impacted. A medical student who had a nursing background added:

*"(. . .) I have been doing the VP course as a tutor. (. . .) I personally make sure that I do not have a group of doctors in the groups and that the nurses are separate, but that I mix them up a bit (. . .). [It is important that] they work side by side (. . .)"* (Interview 8, Student 1)

However, the absence of follow-up courses or further skills training and having it assessment only in the third year of studies were all reasons to consider the workshop inadequate for the first year curriculum.

Another IPE experience mentioned was the two-hour Confidentiality seminar, occurring with law students or with nursing students. Participants attended this seminar in their first year of studies. Most students hinted that the course was not well structured and that students did not mix, so the experience was not really IP. The reason for it being interprofessional was the common topic rather than the interaction between groups.

Five students had additionally chosen to take part in an interprofessional clerkship offered by the University of Bern, consisting of two interprofessional days (first day: nursing students have a shared histology lesson with medical students; second day: nutritional care with student role-play). All students found the IP clerkship very positive. Nursing and clinical clerkships in clinical years, as well as lectures with other professional groups, were also considered IP interventions.

*"I found it so important in my nursing internship that I saw what they actually do, what their tasks are. Because I also noticed from myself that I have a completely wrong picture of what this profession actually is. Because I just thought, a qualified nurse, well. . . and then I saw what they actually do."* (Interview 2, Student 2)

*"We went to lectures for six months with law students. As it was about health law, medical students were also invited. It was very interesting, the law students asked a lot of medical questions which were clear to us, but we didn't know anything about when they mentioned court issues."* (Interview 3, Student 1)

Overall, students welcomed IP courses but were disappointed because of the lack of actual IP (i.e., inadequate setting, disorganized interventions). Medical students felt they had significantly less experience than their IP counterparts.

*"I actually thought [the IPE] was good in the beginning, but in the end we never worked together. (. . .)I think we medical doctors had a lot less experience and it was actually the wrong setting to somehow mix us."* (Interview 8, Student 2)

*"In the intravenous cannulation course, nurses could perform the skill already, because they already had patient contact. And I had zero experience. I profited a lot from them, but I couldn't give them anything in return."* (Interview 5, Student 1)

The IP offer during the Medical course was insufficient: medical students were aware that doctors deal with many other health care professions, and for medical students it would be important to know about other professions' training, roles and responsibilities during the medical

curriculum. Most students did not experience IPE, except for the Intravenous Cannulation course, and one student interviewed had no recollection of any IP interactions during training.

*"We had a couple of IP courses with nursing students during our studies. I thought it was cool, but I think it shouldn't stop there. We will have to deal with so many healthcare groups in the future that it is important to get to know these people during medical studies: what they learn, what they can do and where their limits are. So that we can understand them a little better."* (Interview 5, Student 2)

**Overarching goals of IPE.** Table 6 summarises all the mentioned goals of IPE with the respective quotations. Students named several goals of IPE, segmented into 5 main subcategories:

1. *Profession-linked perspectives*, and *work-oriented learning*: Students were aware that to achieve these goals for application in future daily practice, interactive learning between professional groups was necessary (Table 6, Quote 5).

2. *Improvement of teamwork*: IPE leads to better understanding of the daily routine, work distribution, and duties of other healthcare groups, thus preventing misunderstandings and miscommunication. Enhanced communication through IPE was pointed out as a contributing factor for improved interaction between different professional groups (Table 6, Quote 5).

3. *Reduction of prejudices* in the workplace: Early contact with other healthcare groups could "prevent" the endorsement of stereotypes, and lead to a workplace environment that is open-minded and where there is mutual respect (Table 6, Quote 8).

4. *Enhancement of a patient-centred approach*: IPE implies that patient care is performed collectively, and the patient lies in the centre of care.

5. *Support of workplace wellbeing*: Several students mentioned IPE could create *workplace wellbeing*, particularly by improving social relationships both in and outside work, and by reducing miscommunication, and therefore frustration levels (Table 6, Quote 10).

**Table 6. Subcategory "overarching goals of IPE" elements and representative quotations.**

| Subtheme with explanation | Representative cites (exemplary) from semi-structured interviews |
|---|---|
| *Overarching goals of IPE*<br> • learning together and gaining a more work-oriented perspective<br> • improvement of teamwork<br> • reduction of prejudices<br> • increase in patient-centeredness<br> • improvement of wellbeing in the workplace | Quote 5, Interview 5, Student 1: "(. . .) you have the exchange between different professions very early [during medical school] so you don't come clueless to the hospital later."<br>Quote 6, Interview 1, Student 2: "*You (. . .) become aware of the [roles of team members] and focus on working together.*"<br>Quote 7, Interview 6, Student 3: "*If you have IP communication beforehand, future work with other healthcare groups will be simplified.*"<br>Quote 8, Interview 7, Student 2: "*not letting doctors feel superior to the nurses and correct the stereotype that "nurses only do what we do not want to do cause it´s not good enough or not challenging enough for us"*<br>Quote 9, Interview 2, Student 3: "*I think it is important to learn to appreciate what others do for the patient. During medical school we do not see the whole spectrum [of health care]. Especially the care or the physiotherapy or ergotherapy, too, contribute a lot—and we do not learn about that*"<br>Quote 10, Interview 5, Student 3: "*Also to reduce frustration in the hospital—nurses are frustrated with doctors and the other way around; [IPE] may help*" |

A frequently visited component of IPE was the *enhancement of workplace well-being*. Students were regardful that finding commonalities in different healthcare professions intensifies social relations both inside and outside the workplace, leading to a social benefit. Some students mentioned a financial advantage of IPE, as satisfied staff are more likely to remain in post thus reducing overall costings. Finally, all of the above lead to less medical mistakes, which can increase patient safety.

## b) Barriers to IPE implementation

Issues regarding the *competition with the current medical curriculum*, *the risk of unbalanced learning* and *other dangers* were explored. Students feel they already have an overloaded schedule, so additional IPE interventions could be difficult to implement. They were uncomfortable with being taught by non-doctors because they feared other health care professionals would not be aware of their training or be knowledgeable about their curriculum. The lack of assessment of such activities labels IPE interventions as secondary, superfluous or less relevant. There was an outspoken fear of loss of medical identity, loss of medical specialization (because knowledge is shared), and fear of being less thorough in their own medical curriculum.

> *"You may not get to the level you would need in medical studies if you work with professional groups that are in a specific area that does not have to reach such a high level. And that you may be slowed down a lot in areas."* (Interview 8, Student 1)

> "*It depends on the topic. (. . .) you may have extreme differences in knowledge and personally, I don't think it's so great when I'm somewhere and then I realize that, compared to the others, I don't know anything. I somehow feel stupid and superfluous. I can benefit from the others, but (. . .) it is uncomfortable if you do not participate."* (Interview 8, Student 4*)

On a *course level*, the use of IPE interventions *per se* does not guarantee student interaction. If the IPE experience is not perceived as good by all students, there is a risk that they will consider it unnecessary. The implementation of such activities may be challenging because the content, format and frequency rarely accommodate all students involved. There was a frequently mentioned fear that students would not benefit from the topics due to their diverse backgrounds or varying levels of knowledge on a given subject. Medical students were concerned that topics would be approached too superficially. This could lead to boredom and frustration or create a feeling of unworthiness.

The teaching of competencies outside a given role can lead to a false sense of ability and may have legal consequences (by performing skills outside of set competencies). Additionally, it may enhance prejudices against other health care professions because of single participant's characteristics from each group.

> *"Simply the basic requirements for the [IPE] course were so different that it did not really contribute to bringing these two professional groups closer together, but rather the opposite."* (Interview 1, Student 2)

> *It is difficult to bring the shared content across at a common level so that it is adequate for both groups"* (Interview 6, Student 4)

> *"It is a tightrope walk. IPE is necessary, but it can also be too much."* (Interview 3, Student 4)

Finally, several barriers were mentioned on an *institutional level*: bureaucratic obstacles of combining curricula from different faculties, organizational aspects e.g. lack of infrastructures

to accommodate all students, difficulty in coordinating rotations, time constraints, monetary constraints and deanery or political barriers (resistance to change).

> *"[Barriers include] organization and also coordination with the various training plans. Because we are not learning the same things completely in parallel."* (Interview 6, Student 4)

### c) Expectations of IPE

Ten students agreed that IPE should start as early as the first year of studies. They mentioned several advantages for early IPE introduction which included (1) easier implementation (as students would have similar backgrounds) and (2) the encouragement of early interaction, shared learning and networking, which would contribute to the building of mutual respect from an early stage. Students suggested starting with basic science and other overlapping topics, which could then evolve to clinical interactions later in the curriculum.

> *"And if you start early, you are more sensitive, then you get used to the interprofessional and working together. I think that makes a big difference, even if you are snobbish in the beginning (. . .)."*(Interview 7, Student 3)

Reasons opposed to an early IPE introduction included students being overwhelmed by an overloaded, integrative year; the role of "doctor" not being yet clearly defined and prejudices against other health care professions existing before medical school. On the other hand, eleven students pointed out that the IPE introduction should occur just before or during clinical years (from the third year onwards). For them, it meant a better integration of the IPE content with clinical practice, the previous acquisition of basic clinical knowledge which would facilitate the focus on the IP component, and the broader diversity of activities that could be offered. One student was concerned that such an approach would be too late to prevent the development of prejudices. Five students mentioned it was important to have IPE on a frequent, recurrent basis.

> *"I have the feeling that it is worthwhile, especially later, the more practical it becomes and the more practical things you do, the more it makes sense to integrate IPL. Because the first few years are so theoretical, integration doesn't bring you much."*(Interview 3, Student 3)

> *"But I think that you will probably benefit more from the exchange when you get closer to the clinical semesters. Because [in pre-clinical years] the roles are not yet clearly distributed. Later on the interprofessionality is more noticeable."*(Interview 9, Student 1)

> *"If you just look, whether only earlier or only late, I don't know which would be better. But repeatedly would be good."* (Interview 5, Student 2)

For pre-clinical years, students preferred IPE courses on overlapping topics from basic sciences (e.g., anatomy, physiology, pharmacology. Potential healthcare students to be included were nurses, physiotherapists, midwives and operating room technicians. Courses should be practical (tutorials, case studies, clinical skills trainings, problem-based learning groups, case-based learning) and lectures should be avoided. Other options mentioned included seminars or course days about topics which are relevant to more than one profession or the use of simulation for soft skill and clinical skill training. Some students recommended that such courses should occur during clinical rotations and include other healthcare students. The IP groups should, when possible, be maintained throughout the year to allow for a deeper social interaction.

Students would rather have IPE in smaller groups (4–6 participants, mixed ratio 1:1 or 1:2) to allow for a better interpersonal experience and communication. As for the preferred duration, they felt these should be course blocks of approximately 1–4 hours, entailing a full morning or afternoon. IP courses should have an optional character.

*"If they are smaller groups, if you really have to communicate and interact, then you get to know each other on a more human level and there are many prejudices that can be eliminated."* (Interview 3, Student 2)

*"IPE courses not too often, twice a semester, then increase frequency to once per month towards the end of medical school"* (Interview 6, Student 2)

Students favored regular IPE interventions, with course repetitions. Participants did not agree on an adequate frequency: while some wished for IPE to occur on a weekly, fortnightly or monthly basis, others preferred only once or twice every semester. Some students were concerned about the time it would take to prepare for weekly IPE (e.g., communication) trainings.

Regarding the topic of the IPE intervention, students chose basic science topics for pre-clinical years (including anatomy, biology and patient confidentiality). For clinical years, the main desired interventions included topics like basic life support training, clinical skills training (mostly regarding history and physical examination of organs and systems), handover and rounds, non-technical skills and communication training. Trial (taster) days and areas of shared responsibility (medication errors, hospital hygiene, ethics) were also acknowledged as being useful.

*"I think the focus for IPE is a little bit different. When we are with among medical students, it is often about acquiring knowledge and when it is interdisciplinary, it is more about learning soft skills and how to use them in everyday life."* (Interview 4, Student 1)

## Discussion

This study explored medical students´ attitudes and perceptions towards the main components of IPE in Bern University. The students displayed positive attitudes towards IPE across all study years in individual items, subscales averages and in the global G-IPAS score. This supports findings from a previous Bernese cohort using another interprofessional attitudes scale [25] and reflects similar findings from other countries [26, 27]. Such positive attitudes may be due to a ceiling effect caused by the early exposure to IPE interventions in the Faculty of Medicine of the University of Bern.

Females had significantly more positive attitudes towards interprofessionality in the overall G-IPAS and for the subscales of "teamwork, roles and responsibilities" and "patient-centeredness". Selected studies from Sweden [28, 29], using either the RIPLS or the Jefferson Scale also showed more positive attitudes towards teamwork in females. Others [30] reported a significant effect of gender in the IEPS empathy subscale. No other studies seem to report such a gender effect. Females from these countries (Sweden, Northern Italy, and now Switzerland) may be acculturating in more democratic societies that have a strong egalitarian view of women's position in the workforce. The feeling of being equal to males and having equal work expectations can make such differences more visible. Although many healthcare systems still maintain traditional hierarchical structures and gender roles, they may be transitioning into a more gender-neutral teamwork and patient-centred culture, particularly in central and northern Europe. This is an issue worth exploring in further studies.

Students in pre-clinical years had significantly higher G-IPAS scores. Other studies showed a similar positive attitudes score, both for the healthcare student population in general [6, 31–34] and medicine in particular [6, 35].

One third of students mentioned the importance of the early introduction of IPE in the curriculum, as it facilitated an early interaction and network, contributing to mutual respect and reducing stereotypes. Thus, students can join an interprofessional team without bringing a well-developed "doctor professional identity" [34]. Social Identity Theory [36] supports this: stronger definitions of individual professional roles may lead to intergroup discrimination. Introducing IPE early in the curriculum is likely to have an impact on students' ability to assume their given roles and responsibilities, which is a basic principle of professionalism [37]. Finally, having to learn interprofessional teamwork skills in the workplace in addition to clinical responsibilities and patient care, may increase extraneous cognitive load [38, 39]. Learning these skills may be better served within basic sciences courses, as they provide a more favourable framework for the initiation of IPE [40] Early introduction of IPE would also tackle lower levels of prejudice, promoting more positive attitudes [41].

Factors contributing to this decline in interprofessional attitudes include being more experienced in the healthcare field [32], having previous interprofessional contact [42], having had less positive experiences in IPE [31, 34, 43] and having parents working in healthcare [44]. Although specifically targeted for the Bernese sample, none of these factors showed a significant association with the decline in attitudes. A recent study by Oza et al. [45] applying a regression analysis to a large cohort of medical students, also failed to find such associations with the aforementioned variables. The absence of any association in larger cohorts may be more statistically trustworthy, and the association of these factors in IPE decline should be specifically addressed in higher powered studies.

The decline in students' attitudes towards IPE observed in the quantitative analysis, coupled with 30% of the participants mentioning clear disadvantages of early IPE implementation is worrisome. This is of concern because good relationships with colleagues and patients–likely fostered by IPE–increase patient satisfaction, promote treatment compliance and protect against malpractice claims [46]. Hudson et al. [34] suggested this may be due to the nature of the intervention and how negatively students experienced it. Being taught by non-doctors also reduces medical students´ motivation to participate in IPE interventions [34]. The arguments above, coupled with an underdeveloped professional identity, may have been the reason for the decline. On-going team training may tackle this, as it has been shown to be central in the sustainability of a shared understanding of professional roles [47, 48]. In the present study, students favoured regular IPE to maintain interprofessional proficiency. Both findings reinforce the need to offer health care professional students enough opportunities to interact and learn together from the first year of studies and throughout their careers.

Students had an outspoken fear of loss of medical identity and some showed no positive attitudes towards interprofessionality. Others, despite being at the beginning of their professional career, showed a stereotypical view and regarded interaction between health professions as difficult, which is similar to previous findings [49–52]. Although medical students may lack professional maturity to project the benefits of such IPE experiences, it takes time for a true change in mindset to occur, particularly among professions that have for so long operated independently [53]. Unfortunately, stereotypes formed by professional interaction and societal views on professional roles are not easily modified by educational interactions alone [54]. The introduction of small-group reflections, facilitated by adequate role models, may allow students to remodel their own professional and personal attitude towards patients, to express their moral judgements from their observations of other healthcare professionals' interactions and to share these experiences within a safe learning environment [48]. Such experiences

throughout training programmes may reduce anxieties and fears about future professional collaboration [34].

Students mentioned barriers similar to those noted previously [5, 55], particularly regarding resistance to IPE by students or faculty, difficulty in coordinating coursework and lack of an established framework. Such barriers are able to influence both the outcome as well as the sustainability of an IPE programme [55]. Lawlis [55] also recommends a way to overcome these barriers by means of faculty development plans. Faculty development encourages staff commitment and buy-in, and eases a professional and institutional culture change, in a "bottom up" approach.

The social component of IPE was mentioned as a goal and as an advantage. Students considered the networking beneficial, and by engaging on interprofessional relationships on a personal level, they could learn about each other's curricula in informal settings and even foster friendships. This is a point not frequently explored in the literature. The social aspect repeatedly mentioned in the interviews mirrors many of the components of Social Learning Theory [56]. Learning is also a social and relational process, frequently occurring around authentic and meaningful patient cases [45, 49]. Such findings show that "formal" or planned educational IPE experiences also create "informal" opportunities to socialise and be acquainted on a personal level. These "informal arenas can, therefore, stimulate and set a solid basis for interprofessional collaboration" [54].

All of these observations should be considered in order to offer more authentic interdisciplinary experiences, with the healthcare team and the patient engaging in interprofessional problem-solving activities. Such significant learning interactions have a clear impact on how medical students internalise and approach patient-centeredness [57].

There are limitations to this study: first, the cross-sectional design did not allow for the observation of cohort evolution within their studies and further pre-post analysis. The single-centre design limits the generalization of its conclusions. We tried to overcome this limitation by targeting an adequate sample size, which is one of the largest in IPE literature.

We also cannot assume that our qualitative data can be translated by the simple translation of words, because words and meanings are not equivalent in different languages and language carries a cultural meaning. Although we have used a known approach to translation of our quotes from German to English by two native speakers, our translation may still suffer from misinterpretation and the translated text may break away from the original.

Additionally, we had concerns about the first use of a new scale. Although the G-IPAS was translated and acculturated into German and has shown very solid reliability data and factorial structure, it may not be the appropriate tool for the study´s context. Social desirability bias was also a threat, considering that the G-IPAS was self-reported. Finally, measuring beliefs and attitudes does not indicate true skill proficiency in interprofessional work, and future research should include more ability-oriented measures, aiming for outcomes in levels 3 and 4 of Kirkpatrick´s hierarchy [58].

## Conclusions

Although IPE has only recently been introduced in many healthcare training settings, medical schools and other health professional training institutions have the means to provide opportunities to encourage collaborative interactions early in training. This study´s findings, collected directly from the students, provide valuable insights for the faculty at the University of Bern and for similarly structured universities into the state of IPE in the current programme and potential areas suitable for IPE. They also promote a greater understanding of the difficulties educators and organizations face and encourage discussion about when and how medical

schools should address interprofessional learning. The results from this mixed-methods study demonstrate that medical students are ready for IPE experiences early in their studies.

## Acknowledgments

We would like to thank the Dean´s office of the Medical Faculty of the University of Bern, particularly Dr. Peter Frey, MME for facilitating the distribution of the G-IPAS survey within the University of Bern student´s contacts. Additionally, we thank Dr. Sabine Nabecker for her invaluable help with the first coding of the group interviews. We also thank all the students of the medical faculty who participated in the study.

## Author Contributions

**Conceptualization:** Joana Berger-Estilita, Daniel Stricker, Robert Greif, Sean McAleer.

**Data curation:** Joana Berger-Estilita, Hsin Chiang, Robert Greif.

**Formal analysis:** Joana Berger-Estilita, Hsin Chiang, Daniel Stricker.

**Funding acquisition:** Joana Berger-Estilita, Hsin Chiang.

**Investigation:** Joana Berger-Estilita, Alexander Fuchs, Robert Greif.

**Methodology:** Joana Berger-Estilita, Daniel Stricker, Alexander Fuchs, Robert Greif, Sean McAleer.

**Project administration:** Joana Berger-Estilita, Hsin Chiang, Alexander Fuchs, Robert Greif.

**Resources:** Robert Greif.

**Software:** Daniel Stricker.

**Supervision:** Joana Berger-Estilita, Sean McAleer.

**Validation:** Daniel Stricker, Alexander Fuchs.

**Writing – original draft:** Joana Berger-Estilita, Hsin Chiang, Daniel Stricker, Alexander Fuchs, Robert Greif.

**Writing – review & editing:** Joana Berger-Estilita, Hsin Chiang, Daniel Stricker, Alexander Fuchs, Robert Greif, Sean McAleer.

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
