## [Decision Letter · Decision Letter 0]

25 Aug 2020

PONE-D-20-15351

Attitudes of Medical Students towards Interprofessional Education: A Mixed-methods Study

PLOS ONE

Dear Dr. Berger-Estilita,

Thank you for submitting your manuscript to PLOS ONE. After careful consideration, we feel that it has merit but does not fully meet PLOS ONE’s publication criteria as it currently stands. Therefore, we invite you to submit a revised version of the manuscript that addresses the points raised during the review process.

Please ensure all comments for minor (and some slightly major) feedback changes from the 3 reviewers are included in the next version of the manuscript.

We look forward to receiving your revised manuscript.

Kind regards,

Elisa J. F. Houwink, MD, PhD

Academic Editor

PLOS ONE

Journal Requirements:

'RG is the director of training and education of the European Resuscitation Council, the Task Force Chair Education, Implementation, and Team of ILCOR, and member of the direction of the MME Program of the University of Bern. SM is the Programme Director and Senior Lecturer of the Centre for Medical Education, University of Dundee. The remaining authors report no competing interests. '

a. Please confirm that this does not alter your adherence to all PLOS ONE policies on sharing data and materials, by including the following statement: "This does not alter our adherence to  PLOS ONE policies on sharing data and materials.” (as detailed online in our guide for authors http://journals.plos.org/plosone/s/competing-interests).  If there are restrictions on sharing of data and/or materials, please state these.

Please note that we cannot proceed with consideration of your article until this information has been declared.

3. Please include your tables as part of your main manuscript and remove the individual files.

Please note that supplementary tables should be uploaded as separate "supporting information" files.

4. Your ethics statement must appear in the Methods section of your manuscript. If your ethics statement is written in any section besides the Methods, please move it to the Methods section and delete it from any other section. Please also ensure that your ethics statement is included in your manuscript, as the ethics section of your online submission will not be published alongside your manuscript.

Reviewers' comments:

Reviewer's Responses to Questions

**Comments to the Author**

1. Is the manuscript technically sound, and do the data support the conclusions?

Reviewer #1: Yes

Reviewer #2: Yes

Reviewer #3: Yes

2. Has the statistical analysis been performed appropriately and rigorously? 

Reviewer #1: Yes

Reviewer #2: I Don't Know

Reviewer #3: Yes

3. Have the authors made all data underlying the findings in their manuscript fully available?

Reviewer #1: Yes

Reviewer #2: Yes

Reviewer #3: Yes

4. Is the manuscript presented in an intelligible fashion and written in standard English?

Reviewer #1: Yes

Reviewer #2: Yes

Reviewer #3: Yes

5. Review Comments to the Author

Reviewer #1: Congratulations on a well-designed and conducted study. Just a few clarifying questions.

1. What is the current IPE curriculum? You wrote: "Most frequent IPEs mentioned were the intravenous cannulation course (n=125), the confidentiality seminar (n=98), and the optional interprofessional rotation (n=43)." This should be better explained earlier in the paper. Which respondents and at which level of matriculating year would they have completed that course work and how might it have affected their responses?

2. What was your exclusion criteria? You wrote: "Six-hundred and seventy-seven students replied to the online survey (response rate: 43,7%). After exclusions (n=115), we included 562 questionnaires in the final analysis."

3. A large number of learners had previous healthcare experience. How did this impact your findings?

For clarity's sake for the reader, I recommend that the results section more specifically discuss the tables directly.

Your study is attempting to define when might be the best time to initiate IPE curriculum. Unfortunately, your results do not help clarify that as you do not discuss how stratification by learner level, type, nor prior exposure to IPE curriculum affects that. A sub-analysis of those groups would help elucidate your conclusion that earlier IPE curriculum is better. You mention that earlier learners are more enthusiastic, but do not explore why more advanced students who may or may not have gone through the IPE curriculum did not demonstrate that same enthusiasm.

The qualitative component (Table 4) exposed novel findings including the loss of professional 'uniqueness'. This would be important to explore further as you mention in your discussion.

Reviewer #2: Thank you for the opportunity to review this manuscript.

I have several suggestions listed below. Overall, a spelling and grammar check is needed based on what I presume is the translation to English.

Abstracts states that women "scored higher," but I would suggest rewording as this is an attitudinal survey. It currently seems to imply that women did better.

Background: I would suggest removing the statement "How exactly this occurs is not known..." as you have provided details to support your statement.

In this section, you mention the original development of the IPEC competencies in 2009. There was an important revision in 2016 to note, but I don't know that much information is needed here, as these are common practice currently.

In the demographic characteristics, I would suggest including details and descriptions of the classes noted throughout the paper.

Page 15 Section C- I would suggest rewording the paragraph. It says several advantages with a colon then only one listed. I would combine the information from the first and second sentences. I would also suggest eliminating the sentence "there is no benefit to starting later."

The results section is somewhat confusing. I would suggest eliminating the data that reiterates what is stated in tables 4&5.

Tables 4&5 seem to have a lot of information and make take away from the overall message.

Components of the IPEC report on Page 13 does not seem necessary to be included.

Reviewer #3: Very interesting article which approaches the topic for IPE with a mixed methods approach in a large number of students across the various years that medicine in taught in their institution. This should be commended.

Minor comments

1. too many abbreviations - MS, HCP and parts of the G-IPAS are not needing to abbreviated. I appreciate that they are abbreviated because they are used frequently in the manuscript, but they are not common and add to cognitive load. Please unabbreviate throughout the manuscript.

2. Table 5 - consistency - interview # then student # please.

3. Seeing as there is a strong gender difference in the scoring of the G-IPAS was gender considered in the final model predicting the variance of the overall score? I would have thought there would be an interaction at least within the model. Could the final model also be presented to understand which of the components mostly committed to the variance.

4. Could the authors comment on the frequency of the quotes in each of the interviews. I think this aspect is unclear and I would like a percentage of the main topics across the group to be identified and discussed.

6. PLOS authors have the option to publish the peer review history of their article (what does this mean?). If published, this will include your full peer review and any attached files.

Reviewer #1: No

Reviewer #2: No

Reviewer #3: **Yes: **Kellie A. Charles

---

## [Author Response · Author response to Decision Letter 0]

6 Sep 2020

Reply to Reviewers' comments:

Reviewer #1: Congratulations on a well-designed and conducted study. Just a few clarifying questions.

Comment 1: What is the current IPE curriculum? You wrote: "Most frequent IPEs mentioned were the intravenous cannulation course (n=125), the confidentiality seminar (n=98), and the optional interprofessional rotation (n=43)." This should be better explained earlier in the paper. Which respondents and at which level of matriculating year would they have completed that course work and how might it have affected their responses?

Our reply: Thank you for this comment. We have expanded the description of the IP offer at the University of Bern in the Introduction, which now reads (Page 4, Lines 76-81): “Further interprofessional activities include a compulsory seminar on confidentiality in cooperation with the Bern University of Applied Sciences and the Institute for Medical Education of the University of Bern (UniBe) as well as the compulsory Intravenous Cannulation course, both taught in the first academic year, during which the learning groups and the team of peer tutors are interprofessionally allocated.” 

Comment 2: What was your exclusion criteria? You wrote: "Six-hundred and seventy-seven students replied to the online survey (response rate: 43,7%). After exclusions (n=115), we included 562 questionnaires in the final analysis."

Our reply: Thank you for this remark. We have described our exclusion criteria in more detail and you can now read (Page 9, lines 167-168): “Incomplete questionnaires (n=111) were excluded and 4 students did not report year of studies. We included 562 completed questionnaires in the final analysis.”

Comment 3: A large number of learners had previous healthcare experience. How did this impact your findings?

Our reply: Thank you for your remark. We conducted an independent samples t-test to determine the association between previous healthcare experience and better attitudes towards IP as measured by the G-IPAS. We failed to find a significant difference (Page 14, lines 203-205): “The independent samples t-tests showed no statistically significant difference for previous experience in healthcare and having parents working in the healthcare system.” We added a paragraph in the discussion to explain the impact of these findings and you can now read (Page 26, lines 517-525): “Factors contributing to this decline in interprofessional attitudes include being more experienced in the healthcare field (32), having previous interprofessional contact (42), having had less positive experiences in IPE (31, 34, 43) and having parents working in healthcare (44). Although specifically targeted for the Bernese sample, none of these factors showed a significant association with the decline in attitudes. A recent study by Oza et al. (45) applying a regression analysis to a large cohort of medical students, also failed to find such associations with the aforementioned variables. The absence of any association in larger cohorts may be more statistically trustworthy, and the association of these factors in IPE decline should be specifically addressed in higher powered studies.”

Comment 4: For clarity's sake for the reader, I recommend that the results section more specifically discuss the tables directly.

Our reply: Thank you. We have introduced subheadings in the results section (“Quantitative analysis” and “Qualitative analysis”) and have taken on board the suggestion from Reviewer #2, Comments 7 and 8, and have introduced relevant citations in the results section, significantly reducing the information in Tables 4 and 5. We hope that this will enhance readability of the qualitative analysis results. 

Comment 5: Your study is attempting to define when might be the best time to initiate IPE curriculum. Unfortunately, your results do not help clarify that as you do not discuss how stratification by learner level, type, nor prior exposure to IPE curriculum affects that. A sub-analysis of those groups would help elucidate your conclusion that earlier IPE curriculum is better. You mention that earlier learners are more enthusiastic, but do not explore why more advanced students who may or may not have gone through the IPE curriculum did not demonstrate that same enthusiasm.

Our reply: Please refer to the reply to comment 3. Page 26, line 526 to Page 27, line 538 looks at further reasons for a decline in motivation.

Comment 6: The qualitative component (Table 4) exposed novel findings including the loss of professional 'uniqueness'. This would be important to explore further as you mention in your discussion.

Our reply: Thank you for this insightful comment. We have commented on the loss of professional identity in Page 27, lines 539-552. 

Reviewer #2: Thank you for the opportunity to review this manuscript.

Comment 1: I have several suggestions listed below. Overall, a spelling and grammar check is needed based on what I presume is the translation to English.

Our reply: Thank you for this comment. We are aware of the translation issues in qualitative, interdisciplinary research. Such translation issues arise more and more frequently, as the most highly ranked international academic journals are mostly published in English. We, as non-English-speaking academics, are confronted with the challenge of translating our research results into English, and such translation processes may come with additional language challenges and issues. Such language and translation issues are particularly important in qualitative interview-based research. Interviews often aim to unveil interviewees’ subjective experiences, usually expressed in their source language – that is, a language other than English. This is what happened in our case. In qualitative, interview-based accounting studies, translation problems primarily materialize in direct quotations used in the manuscripts reporting on research results. For such quotations, researchers need to translate material directly from non-English interviews into the English language. We see direct quotations from interviews as an opportunity to achieve credibility and authenticity in qualitative analysis. Additionally, a proper translation of quotations from non-English interviews may not be easy to achieve, since the original meaning of the quotations needs to be preserved. In order to cope with this limitation, we have used the concept of creating equivalent translational structures, a core task in translation processes (Enzenhofer and Resch, 2011). This concept refers to the establishment of similarity between the source and the target languages at the textual level. This functionalist approach of translation, performed by one of the authors (SM), aims for the achievement of a translation initiator’s needs (Schäffner, 2009), which involves ensuring the target text is understandable for an end user. Consequently, the translated text may break away from the original text. We also introduced a paragraph on this topic both in the Methods section and in the limitations section of the manuscript. You can now read:

Methods, Page 7, Line 157 to Page 8, Line 162: “Direct quotations from the interviews were translated into English using a functionalist approach of creation of equivalent translation structures as described by Enzenhofer and Resch (23). One author (HC, German-speaking) translated the citations from German to English ipsis verbis with the aid of an online tool (Google Translate®). The second author (SM, English-speaking), performed changes to ensure that the target text could be understood by the reader.”

Limitations, Page 29, Lines 578-582: “We also cannot assume that our qualitative data can be translated by the simple translation of words, because words and meanings are not equivalent in different languages and language carries a cultural meaning. Although we have used a known approach to translation of our quotes from German to English by two native speakers, our translation may still suffer from misinterpretation and the translated text may break away from the original.”

Comment 2: Abstracts states that women "scored higher," but I would suggest rewording as this is an attitudinal survey. It currently seems to imply that women did better.

Our reply: Thank you. We reworded as suggested and it reads now: “showed better attitudes” (Page 2, Line 36.

Comment 3: Background: I would suggest removing the statement "How exactly this occurs is not known..." as you have provided details to support your statement.

Our reply: Thank you. We removed the statement as suggested.

Comment 4: In this section, you mention the original development of the IPEC competencies in 2009. There was an important revision in 2016 to note, but I don't know that much information is needed here, as these are common practice currently.

Our reply: Thank you. We have deleted the statement regarding the revision of the IPEC report. You can now read (Page 3, Lines 54-56): “The Interprofessional Collaborative Practice (IPEC) outlines IPE’s core competencies that concentrate on four main domains: Ethics & Values, Roles & Responsibilities, IP Communication and Teamwork.(4)”. We also updated the reference to the 2016 update of the IPEC report. 

Comment 5: In the demographic characteristics, I would suggest including details and descriptions of the classes noted throughout the paper.

Our reply: Thank you for this comment. We are not totally sure what the reviewer means with his/her comment but we guess it is about the different experiences of the students. Therefore, we have expanded the qualitative analysis results section and added student´s comments and descriptions of their experiences with specific interprofessional activities. You can now read:

Page 17, Lines 266-270: “However, most students realised that nursing students already had the given competency and were bored/frustrated during the workshop. Some medical students observed other peers having discriminating attitudes towards nursing students. Most were unhappy to be in a workshop where they knew less than their nursing counterparts and could not contribute to any exchange in knowledge.”

Page 17-18, Lines 294-307: “However, the absence of follow-up courses or further skills training and having it assessment only in the third year of studies were all reasons to consider the workshop inadequate for the first year curriculum.

Another IPE experience mentioned was the two-hour Confidentiality seminar, occurring with law students or with nursing students. Participants attended this seminar in their first year of studies. Most students hinted that the course was not well structured and that students did not mix, so the experience was not really IP. The reason for it being interprofessional was the common topic rather than the interaction between groups. 

Five students had additionally chosen to take part in an interprofessional clerkship offered by the UniBern, consisting of two interprofessional days (first day: nursing students have a shared histology lesson with medical students; second day: nutritional care with student role-play). All students found the IP clerkship very positive. Nursing and clinical clerkships in clinical years, as well as lectures with other professional groups, were also considered IP interventions.”

Comment 6: Page 15 Section C- I would suggest rewording the paragraph. It says several advantages with a colon then only one listed. I would combine the information from the first and second sentences. I would also suggest eliminating the sentence "there is no benefit to starting later."

Our reply: We are sorry if our description was more dubious than intended. We have corrected the signalled sentence and one can now read (Page 22 , Lines 414 - 419: “Ten students agreed that IPE should start as early as the first year of studies. They mentioned advantages for early IPE introduction which included (1) easier to implementation (as students would have similar backgrounds) and (2) the encouragement of early interaction, shared learning and networking, which would contribute to the building of mutual respect from an early stage. Students suggested starting with basic science and other overlapping topics, which could then evolve to clinical interactions later in the curriculum.”

Comment 7: The results section is somewhat confusing. I would suggest eliminating the data that reiterates what is stated in tables 4&5.

Our reply: Please refer to Comment 4 from Reviewer #1.

Comment 8: Tables 4&5 seem to have a lot of information and make take away from the overall message.

Our reply: Please refer to Comment 4 from Reviewer #1.

Comment 9: Components of the IPEC report on Page 13 does not seem necessary to be included.

Our reply: Thank you. We removed them as suggested.

Reviewer #3: Very interesting article which approaches the topic for IPE with a mixed methods approach in a large number of students across the various years that medicine in taught in their institution. This should be commended.

Minor comments

Comment 1: too many abbreviations - MS, HCP and parts of the G-IPAS are not needing to abbreviated. I appreciate that they are abbreviated because they are used frequently in the manuscript, but they are not common and add to cognitive load. Please unabbreviate throughout the manuscript.

Our reply: We would like to thank the reviewer for this comment. We edited the manuscript as suggested.

Comment 2: Table 5 - consistency - interview # then student # please.

Our reply: Thank you for pointing this out. We corrected the tables accordingly.

Comment 3: Seeing as there is a strong gender difference in the scoring of the G-IPAS was gender considered in the final model predicting the variance of the overall score? I would have thought there would be an interaction at least within the model. Could the final model also be presented to understand which of the components mostly committed to the variance.

Our reply: Thank you for your comment. Gender differences were indeed found across many variables and especially also for the overall G-IPAS mean score. However, we have found gender differences only as main effects and not as interaction effects. Gender plays a role but the variable does not interact with study year in the ANOVA (although the alpha error was 0.068 and barely missed significance) nor does gender interact with the variable clinical years, where we found higher G-IPAS overall scores for students in pre-clinical years (1-3) compared to those in clinical years (4-6). For the latter case, we initially did not take gender into the model since it would not have added additional information. Just to be on the safe side, we expanded the last analysis and did an ANOVA including gender as factor and pre-clinical vs clinical years as second between group factors. We found no interaction between the two factors with regard to the overall G-IPAS score (p=0.573), the main effects gender, as reported in the first ANOVA, and clinical years, as reported from the t-test, reached significance. However, in order to give a better insight into the data, we have added the F-values and the partial Eta-squared for the effects of the ANOVA.

Comment 4: Could the authors comment on the frequency of the quotes in each of the interviews. I think this aspect is unclear and I would like a percentage of the main topics across the group to be identified and discussed.

Our reply: Thank you for this important comment. We have added the frequency of quote themes in the qualitative analysis section (Table 4). One can now read (Page 14, Lines 216-218): Table 4 depicts the overall frequency of the quotes in each of the interviews. There were similar distributions of codes across genders and years of study.

Table 4: Coding frequency across all interviews

 Frequency (n) Percentage (%)

Participant´s age 31 3.33

Participant´s year of studies 31 3.33

Participant´s previous work experience 15 1.61

Participant´s ties with healthcare 41 4.40

Comments on filling the GIPAS form 25 2.68

Definition of IPE 44 4.72

Goals of IPE 48 5.15

Advantages of IPE 112 12.02

Disadvantages of IPE 101 10.84

Examples of IPE during medical course 96 10.30

Attitudes towards IPE 63 6.76

Attitudes: Absence of IPE 34 3.65

Examples of wished for interventions 70 7.51

Desired format of the IPE course 71 7.62

Desired Year of studies for IPE 92 9.87

Desired Frequency of IPE 50 5.36

Ideal group size for IPE interventions 8 0.86

Total number of coded citations 932 100

Journal Requirements:

1. Please ensure that your manuscript meets PLOS ONE's style requirements, including those for file naming. The PLOS ONE style templates can be found at https://journals.plos.org/plosone/s/file?id=wjVg/PLOSOne_formatting_sample_main_body.pdf and https://journals.plos.org/plosone/s/file?id=ba62/PLOSOne_formatting_sample_title_authors_affiliations.pdf - DONE

2. Thank you for stating the following in the Competing Interests section: 'RG is the director of training and education of the European Resuscitation Council, the Task Force Chair Education, Implementation, and Team of ILCOR, and member of the direction of the MME Program of the University of Bern. SM is the Programme Director and Senior Lecturer of the Centre for Medical Education, University of Dundee. The remaining authors report no competing interests. ' 

a. Please confirm that this does not alter your adherence to all PLOS ONE policies on sharing data and materials, by including the following statement: "This does not alter our adherence to PLOS ONE policies on sharing data and materials.” (as detailed online in our guide for authors http://journals.plos.org/plosone/s/competing-interests). If there are restrictions on sharing of data and/or materials, please state these. Please note that we cannot proceed with consideration of your article until this information has been declared. - - DONE

b. Please include your updated Competing Interests statement in your cover letter; we will change the online submission form on your behalf. Please know it is PLOS ONE policy for corresponding authors to declare, on behalf of all authors, all potential competing interests for the purposes of transparency. PLOS defines a competing interest as anything that interferes with, or could reasonably be perceived as interfering with, the full and objective presentation, peer review, editorial decision-making, or publication of research or non-research articles submitted to one of the journals. Competing interests can be financial or non-financial, professional, or personal. Competing interests can arise in relationship to an organization or another person. Please follow this link to our website for more details on competing interests:http://journals.plos.org/plosone/s/competing-interests - DONE

3. Please include your tables as part of your main manuscript and remove the individual files. Please note that supplementary tables should be uploaded as separate "supporting information" files. - DONE

4. Your ethics statement must appear in the Methods section of your manuscript. If your ethics statement is written in any section besides the Methods, please move it to the Methods section and delete it from any other section. Please also ensure that your ethics statement is included in your manuscript, as the ethics section of your online submission will not be published alongside your manuscript. - DONE

---

## [Editor Report · Decision Letter 1]

5 Oct 2020

Attitudes of Medical Students towards Interprofessional Education: A Mixed-methods Study

PONE-D-20-15351R1

Dear Dr. Berger-Estilita,

We’re pleased to inform you that your manuscript has been judged scientifically suitable for publication and will be formally accepted for publication once it meets all outstanding technical requirements.

Kind regards,

Elisa J. F. Houwink, MD, PhD

Academic Editor

PLOS ONE
---

## [Editor Report · Acceptance letter]

12 Oct 2020

PONE-D-20-15351R1 

Attitudes of medical students towards interprofessional education: A mixed-methods study 

Dear Dr. Berger-Estilita:

I'm pleased to inform you that your manuscript has been deemed suitable for publication in PLOS ONE. Congratulations! Your manuscript is now with our production department. 

Kind regards, 

on behalf of

Dr. Elisa J. F. Houwink 

Academic Editor

PLOS ONE